# A Novel Strategy for the Assessment of Radon Risk Based on Indicators

**DOI:** 10.3390/ijerph18158089

**Published:** 2021-07-30

**Authors:** Simona Mancini, Martins Vilnitis, Michele Guida

**Affiliations:** 1Department of Computer Engineering, Electrical Engineering and Applied Mathematics (DIEM), Laboratory “Ambients and Radiations (Amb.Ra.)”, University of Salerno, 84084 Fisciano, Italy; miguida@unisa.it; 2Institute of Construction Technology, Faculty of Civil Engineering, Riga Technical University, 1658 Riga, Latvia; Martins.Vilnitis@rtu.lv

**Keywords:** radon assessment, indoor radon, radon in building materials, environmental radon, radon exposure, health risk of radon, radon and public health

## Abstract

Among the physical pollutants affecting indoor air, the radioactive gas radon may turn out to be the most hazardous. Health effects related to radon exposure have been investigated for several decades, providing major scientific evidence to conclude that chronic exposures can cause lung cancer. Additionally, an association with other diseases, such as leukemia and cancers of the extra-thoracic airways, has been advanced. The implementation of a strategy to reduce the exposure of the population and minimize the health risk, according to the European Directive 59/2013/Euratom on ionizing radiations, is a new challenge in public health management. Starting from an understanding of the general state-of-the-art, a critical analysis of existing approaches has been conducted, identifying strengths and weaknesses. Then, a strategy for assessing the radon exposure of the general population, in a new comprehensive way, is proposed. It identifies three main areas of intervention and provides a list of hazard indicators and operative solutions to control human exposure. The strategy has been conceived to provide a supporting tool to authorities in the introduction of effective measures to assess population health risks due to radon exposure.

## 1. Introduction

Air pollution causes serious harm to human health [1]. Many epidemiological studies, over the decades, have demonstrated how human health can be strongly influenced by environmental factors, including exposure to physical, chemical, biological, and radiological contaminants in the environment. Protecting human health from environmental pollutants is an urgent mission for public health authorities. Thus, the assessment and control of health risk from air pollution plays a very crucial role in any realistic roadmap for achieving public health and well-being. Generally, air pollution health risk assessment (AP-HRA) [2] mainly focuses on outdoor air, such as the monitoring of chemical compounds and PM pollutants, especially in urban areas. However, AP-HRA should be more strongly extended to indoor air, too, since air pollution in confined spaces is likewise one of the leading risk factors for deaths globally.

Commonly, HRA in indoor air principally pays attention to chemical and biological agents or ergonomics, lighting, and microclimate factors, since they more frequently affect closed environments, often neglecting one of the most hazardous physical agents: the carcinogenic naturally occurring radioactive gas radon (222Rn) [3,4]. It is well established, in fact, that there is a clear connection between indoor exposure, by inhalation, to radon and the incidence of lung cancer [3]. Furthermore, positive correlations between radon and other harmful indoor pollutants are emerging from the most recent and accurate investigations, as it has been observed in the case of phthalates for children [5].

Radon short-lived alpha-emitting daughter progeny (218Po, 214Pb and 214Po) (Radon Daughter Progeny, RDP), once inhaled, can be deposited in the lungs and the respiratory tract in general and thereby be the cause of high doses from the high linear energy transfer (LET) alpha particle radiation emitted, determining a so-called ‘internal exposure’.

Due to its carcinogenicity, radon is a public health concern, and its monitoring in indoor environments for protection against the dangers arising from exposure to ionizing radiation is recommended by national and international authorities [6]. Thus, a widespread interest has grown in recent decades about an understanding of the complex phenomena of indoor radon accumulation and in the development of policies and methods to monitor and reduce human exposure to ‘safety’ levels. Interest has grown in recent years after the publication of the last European Directive 59/2013/Euratom [7] which, according to new epidemiological studies, introduces more stringent measures for protection, including for private households, and extends control to all possible sources of radon in a confined environment. Therefore, the implementation of an effective strategy for HRA to radon exposure is an urgent challenge in public health management.

Systematic reviews on the matter, in the past years, have shown a lack of harmonized methodologies to assess this risk and the absence of a general strategy to be implemented in different scenarios [8]. To be successful, a strategy needs to be comprehensive and scientifically sound and able to be implemented in the specific conditions of the country, including, for example, outdoor climate, building design, types of building material used, and knowledge and behavior patterns of the occupants.

Therefore, the scope of this paper is to propose a novel general strategy for the control of radon exposure at the national and regional level, according to the most recent regulatory advances and scientific results gathered over the past decades. The proposed approach is based on the definition of control indicators for each potential source of the hazard, ranked in classes according to the severity of the impact. This preliminary proposal aims to constitute the basis for the future development of a health risk model (HRM) based on the calculation of an exposure score (ES), able to determine and predict the global exposure of the population to indoor radon [9].

## 2. The Framework

The assessment and control of indoor radon is a public health risk management matter [10,11]. The proper modeling of a general strategy for the managing of a public health matter firstly requires an accurate definition of the global framework in order to understand who the target population is, where the risk can occur, what the effects are on health, and the main determinants responsible for an increasing risk [12].

Many studies have been carried out on the detection and interactions among the determinants that outline a general public health risk. It is largely accepted that the framework is based on three basic sets (Figure 1) of population health determinants [13]:(a)biology and genetic endowment;(b)environment and occupation;(c)social and behavioral factors.

Applying the conceptualization of Figure 1 to depict the issue, it is possible to understand the relationships and interactions among radon and the different determinants.

(a) Biology and genetic endowment. Feeble biological structure and genetic vulnerability make some people more susceptible to some environmental stressors than others. Susceptibility to radon varies according to the age, sex, and habits. For example, children can be more affected than adults due to higher respiration rates, smokers or past smokers can be exposed to a synergistic effect of radon-tobacco smoke, and men seem to have a higher baseline risk than women, in terms of developing lung or blood cancer [14,15,16].

(b) Environmental and occupational. The radioactive gas radon, decay product of radium (^226^Ra), member of the uranium series (^238^U), is abundantly and ubiquitously naturally produced in the earth’s crust. Once released from soil pores, due to its half-life (3.82 days), it can migrate within rocky materials, where it has been produced (emanation process) and transported across the near-surface soils by fluid carriers (as water, air, CO2, CH4) through advective and diffusive mechanisms, favored by the soil mechanical characteristics (porosity, permeability, and structure) and the environmental conditions. After reaching the external atmosphere (exhalation process) it can enter and accumulate in closed spaces under particular conditions (poor ventilation, presence of cracks in the basement, etc.). Therefore, radon diffuses and degrades in the environment at different speeds in different geological, seasonal, and meteorological conditions.

As naturally present in the soil, capable of dissolving in water used for human consumption and accumulating in a closed environment, it constitutes an environmental and occupational problem at the same time.

(c) Social and behavioral. There are clear socio-economic differences in radon-related awareness, risk perception, and behavior between rural and urban areas. Lifestyle is different, too. Adults in urban areas spend about 93.75% of their time indoors, either working, studying, playing, or maintaining a sedentary lifestyle, mainly in the long winter season in the Nordic countries, whereas an increased mobility experienced in summertime decreases the extent of exposure [17].

Regarding the interactions with other determinants, referring to Figure 1, the environment-occupational determinant is connected to the others with the following interactions:Biology-environment. Radon exposure occurs mostly in old, damaged houses where cracks in the basement and walls represent entry points of radon from soil. Oftentimes, these houses are poorly ventilated, favoring gas accumulation, with quite a lot of dust, aerosols, and combustion by-products, which can attract radon daughter progeny (RDP), which, once inhaled, settles in the lung mucosa. In this example of environmental context, lung cancer susceptibility is related to the individual functional capability to signal, via ubiquitination processes, DNA damage and to repair radiation-induced double-strand breaks. Therefore, genetic factors are significant contributors to the pathogenesis of lung cancer due to the exposure to the radon pollutant.Environment-social. Individual behavior to risk responses such as smoking, a sedentary life, closing windows for personal protection against burglary or smog, or the habit to sleep in the basement to avoid traffic noise may increase the level of exposure, as very poorly ventilated environments contribute to radon accumulation, and smoking has a synergistic effect. Therefore, social habits could determine an amplification of the exposure.

All of this assumed, a strategy for the assessment and control of the radon hazard should aim to estimate the risks of current and future exposure, as well as changes that may result from modifications of the conditions, and be able to correctly assess:(i)the amount of radon present in the air (i.e., the activity concentration);(ii)the amount of exposure of the targeted population;(iii)how harmful the concentration is for human health, i.e., the resulting health risks to the exposed population [2].

In the next section, the main methodological steps for a strategy that models, in a more complete way, the resulting health risks to the exposed population from all the radon sources are described.

## 3. The Method

In the management of a risk (Figure 2), in compliance with the ISO standard 31000:2018 [18], it is important to identify the sources and the consequent effects in order to determine the risk priority.

Among the most common techniques for the analysis of exposure to polluting agents, the definition of indexes represents a very useful tool to easily describe the quality of the environment. Based on experimental measurements, the approach using indicators realizes a quantitative and qualitative picture of the ‘health status’ of the environment. For this reason, it is considered to be one of the most transparent and efficient tools to support decisions and actions for competent control authorities and to communicate with public opinion [19].

Therefore, in the management of the health risk due to exposure to environmental radon, it is first important to categorize the main sources of radon influencing the global indoor accumulation dynamics. Then, methods and indicators to control the risk of exposure must be defined.

The dynamics of indoor radon accumulation is a complex phenomenon [21], determined by the interaction of many parameters, which, in the most general situations, can be time- and space-dependent (Figure 3). Therefore, it is more practical to operate a simplification focusing only on the major influencing sources, processes, and factors.

In this way, a general simplified scheme (Figure 4) was realized by one of the authors [20]. As shown in the figure, radon can reach an indoor environment by coming mainly from the geogenic compartment (soil) and the anthropogenic compartment (building materials, water, and gas supplies). Stratigraphy and geological and hydrogeological features are the most relevant factors influencing high radon activity concentrations in the soil. The structural and plant features of the building influence the accumulation of indoor radon. Regarding radon in water, for simplicity, we are neglecting the risk due to ingestion (which is not so universally established by the scientific community) and considering only that due to inhalation.

The second step is to identify practical tools to assess the probability of the risk. Indicators are valid instruments for this purpose. Typically grouped into ranges, according to defined classes describing the magnitude of the impact, indicators easily communicate the environmental conditions, since according to the severity of the impact, a descriptor as a colour code or a standardized public health advisory is assigned.

In this paper, the development of three indicators describing the grade of the potential hazard from each source of radon in an indoor environment is proposed. The work is conceived as a useful tool for authorities to identify buildings requiring intervention to improve the sustainability of the building’s behaviour to radon, to support urban planners in the identification of radon-prone areas [22], to support professionals in the design of new sustainable structures, etc.

## 4. Results and Discussion

Even if conceptually simple, the development of indicators requires facing some problems. First of all, the definition is not easily internationally common, since it varies reflecting the governmental directions in terms of respecting national reference or action level values and the quality standards adopted [23]. Then, the choice of a unique index requires the identification and assessment of a number of variables, calculation methods, and the definition of different categories of risk. In this context, after having analysed the procedures, standards, and indicators used in different countries or proposed in the scientific literature, the main ones were selected by the authors and summarized in Table 1.

Then, the pros and cons of the selected indexes (Table 2) were identified, and new revised ones were proposed (Table 3). For water, no adjustment has been considered for the calculation of the Cw parameter since it is, by itself, sufficiently exhaustive on the basis of the general knowledge of the contribution of de-gassed radon from water to the indoor environment. Therefore, the work has been restricted only to the definition of Cw.

Instead, for soil and building materials, adjustments or new calculations have been proposed (Table 3). As reported in Table 3, the new proposed indexes attempt to overcome the limitations (cons) of the previous ones.

For the soil, one of the most important limitations of the GRI, reported by the authors themselves [24], concerns the predictive capacity being not as expected, despite the accuracy of the calculation. This limitation results in thinking that in the assessment of a complex phenomenon, some simplification could be considered in order to facilitate applications. Since, despite the accuracy, the predictive capacity is not much better than that resulting from a simplified approach, here, a more simplified index is proposed on the basis of a previous investigation done by the authors [26]. Based on field measurements, the easy to perform and technically simple Geogenic Rad-Campania (GRC) approach enables one to redact quite accurate cartographies of the radon potential from soils, from small to large scales. This method has already been successfully applied at provincial and local scales [26,27]. The level of risk is expressed in classes, from very low to very high, as a combination of the level of exposure and hazard. Then, the GRC index is defined as the ratio between the radon activity concentration measured in the soil-gas and the smaller limit of the radon activity concentration characterizing the very high class (i.e., corresponding to 500,000 Bq/m^3^ as defined in [26]). This definition ensures that 1 turns out to be the maximum reference level of risk.

An important remark concerns the choice of the reference value. In fact, depending on its value, the indices could be greater or less than 1 (more or less than the reference value). In this way, a well-defined global quantity, easily and clearly interpreted, can be obtained. The class of the index is instead described using a table of colours. This important step moves from the certainty that a unique general index would be easily understood by the public to a calculation in a simpler manner using reasonable assumptions and descriptors (Table 4). Then, according to the class, actions could be mandatorily required. Actions regard the mandatory application of mitigation systems inside the building foundations (application of radon barriers, pumps and sumps, etc.).

Of course, the practical potentiality of a radon potential cartography lies in its capability to identify, in each province, the districts with a high susceptibility to radon exhalation from soil and, for each district, the portions of the municipal territory which exhibit or could exhibit high radon concentrations, with the possibility of imposing preventive actions on buildings. However, in addition to being an important tool for strategic urban planning, it could be combined with the other tools defined here to assess the indoor radon health risk.

Regarding the construction of an index for the assessment of the risk from BM, the principal limitation is related to the fact that the radiological characterization of the materials, in many countries, contemplates only the γ exposure. Moreover, its calculation in different countries [23] refers to different models and different standards. In particular, concerning the different formulas used internationally, the index is defined in terms of radium content. From a radioprotection point of view, the content of radium represents, of course, an index of the potential hazard, but, in practical terms, it could be more suitable to also refer to another parameter: the radon exhalation rate. If the radium yield gives a direct measure of the potential hazard (the larger the yield, the higher the probability to release radon, of course), the exhalation rate, instead, also better represents a sort of ‘efficacy’ of the hazard. The specific exhalation rate is related to the radon flux emitted from the building material per mass unit (per surface unit). It is a reference parameter, generally used in scientific literature, to identify the contribution of the building materials to indoor radon. The radon exhalation rate of a building material is influenced not only by the radium content but also by porosity, water content, permeability, emanation power or fraction, surface preparation, and covering. For this reason, considering it could be more representative of the real hazard related to the alpha exposure. By such measurements, the exhalation rate can be calculated by referring to the absolute dimensions (the amount of material) as well as the real shape (surface-to-volume ratio) of the sample and can complete the ‘technical radiological information sheet’ of the material sample. In this direction, an index was proposed by Trevisi et al. in 2013 [23], but the cons related to this proposal concern the calculation of the exhalation rate that would be better determined not by means of radium content but by means of appropriate measurements through accumulation chambers [28]. This solution enables having a more accurate knowledge of the effective radon exhalated from samples made of multiple layers of different building materials, for example.

The calculation of the index, also in this case, comes from direct measurements of the samples and then the application of a formula according to which the index varies from 0 up to values >1. Since 1, also in this case, is the reference limit, some restrictions could occur in the utilization of the building materials for indoor use. Therefore, we can build, as in the case of soil, a class of levels by color to indicate the quality of the BM for indoor use (Table 5). Then, according to the class, restrictions in the quantity or in the use could be mandatory. In this direction, the application of a voluntary label to certify building materials as ‘radon tested’ could be important in the context of the promotion of sustainability in construction.

Then, to determine the total health impact to an indoor environment due to the presence of radon sources from soil and building materials, a combination of the two above introduced indexes is proposed as follows:I_tot_ = (aI_S_) (bI_BM_)
where a and b are dimensionless weights amplifying or reducing the contribution of the BM and soil sources in the total indoor radon accumulation phenomenon.

The utility of a global index lies in the fact that it could be possible to choose one building material rather than another according to the class of the soil, in order not to contribute further to the accumulated indoor radon. Another use of this index is related to the possibility of easily identifying the buildings more susceptible to high indoor radon according to the structural features of the building and the geological characteristics of the soil underneath.

All of this assumed, the new indicators for a sort of early warning analysis having been discussed, revised, and proposed, the methodology is graphically represented in Figure 5. 

In particular, three areas of interventions have been defined: soil, building materials, and indoor environment. Indeed, since indoor radon accumulation is not only due to the transport, driven by pressure differences, from the terrain to the building through the basement, but it is also due to the direct exhalation from building materials, existing buildings should be renovated through opportune interventions of mitigation or equipped with a real-time radon sensor network system. The installation in buildings of a radon sensor system able to launch an instant alert in the case of exceeding concentrations and to report the average measured levels represents a cost-effective solution to prevent the risk of excess exposure. The advantages of this solution are related to reduced environmental costs (compared with standard technologies), an instant alert system in the case of exceeding concentrations, the provision of periodic reports, more reliable measurements than standard solutions (such as the use of passive dosimeters, which does not avoid the tampering with measurements, especially in workplaces and public places), and the prompt execution of mitigation action through the activation of HVAC systems only in case of exceeding radon levels, with obvious economic savings. Furthermore, it is able to provide to the occupants with information about the real radon levels of exposure well in advance with respect to the annual duration of a standard monitoring prescribed by the valid legislation. In this way, public awareness with respect to the radon issue can be largely improved, making people aware of the phenomenon.

The control of radioactivity induced by BM is crucial for new buildings or restored ones. Since modern society promotes a new philosophy in building construction based on the concept of ‘sustainability’, many international voluntary labels have been created with the purpose to certify the non-toxicity of building materials and of indoor air and already consider the control of natural radioactivity. All the above-mentioned protocols and voluntary labels refer to the calculation of the gamma dose due to building materials, ignoring the alpha dose, which is more dangerous than gamma for human health because it is related to ‘internal’ exposure. Therefore, the future goal is the proposal of a label accompanying the different materials used in construction, similar to the certificate of origin and tracking accompanying food found on department stores’ shelves, capable of exhibiting in a simple, understandable, and transparent way the potential hazard associated with the exposure to radon exhaled by these materials, integrating the standard control on the gamma dose required by the regulations. The innovation of this idea consists of the fact that the volunteer label provides for target businesses operating in the field a single information and communication tool, essential for enhancing the features of the bio-sustainability of its products, and a reliable and accredited safety protection and public health safeguard, thus increasing the satisfaction and trust of the customer and the end user.

The idea of developing a certificate measuring radon emission, in terms of evaluating the human health risk, in reference to current regulations, has to be supported by the definition of standardized techniques and methods in order to publish the label. First of all, the measurement should be realized on a sample of standardized dimensions of building materials for indoor environment use. Each sample should be representative of the different raw materials and origins. The exhalation rate of radon from building materials can be determined by studying the growth of radon activity concentrations in closed vessels containing samples of them. Indeed, among all the possible measurement techniques, the radon chamber technique is simple and low cost and widely used to determine the exhalation rates [29].

Regarding the control of the hazard coming from soil, radon potential maps are an efficient basic tool for territorial planning.

The practical potentiality of a radon potential cartography lies in its capability to identify in each province the districts with a high susceptibility to radon exhalation from soil and, for each district, the portions of municipal territory which exhibit or could exhibit high radon concentrations, with the possibility to program, in a specific way, a monitoring campaign or to impose preventive action on buildings included in those areas. The recent European Directive [7] also provides for these cases the possibility of including in the national building codes the obligation to already adopt preventive measures in the construction phase of new buildings.

To integrate all these technical solutions, a global radon certification of buildings could definitely be introduced.

In some highly developed countries, real estate market purchase and sale transactions of homes require a certification concerning typical indoor radon levels and the adoption or not of mitigation remedies [30]. As is already the case for energy consumption and energy efficiency mitigation actions, the promotion of radon certifications conducted by independent and qualified experts and subsidies to cover up to half the costs of the mitigation for the homeowners could be a boost for general control.

Certification software should be based on algorithms modeling the indoor radon dynamics. Then, a classification of houses in ‘Radon classes’, from high to low, and the design of a structural and technical solution to prevent radon entry into buildings should be provided. The conceived software [31] should be able to simulate typical concentrations in the detected houses. Regarding this prevision, comparing the results with the defined classes of concentrations, a classification of the houses in low, medium, and high radon potential could be produced based on the certification. A possible measure of action for the reduction of radon entry and exposure can be included in the certificate with the simulation of future indoor radon concentration after the configured scenario. A software of this kind is practical and effective, as opposed to the performance of operative direct measurements in the buildings integrated in a year and to be repeated periodically. This kind of certification can be added to traditional software for the energy certification of buildings and be CAD-based, because it requires the introduction of some data already required for the energy classification.

All the proposed solutions identify the radon potential from soil, building materials, and indoors ranked in classes from low to high based on the value of the hazard indicated. In this way, they not only give a practical idea of the impact of the radon potential but also constitute the basis of a qualitative measure of the risk, which can be defined through different methods, such as the risk matrix or indicator-based approach, etc.

## 5. Conclusions

The development of methodologies for early warning analysis, control of the risk, and optimization of the solutions is an important task in the management of every issue. From food safety to energy consumption [32,33], the construction of the right process approach is very important for any effective, efficient, and successful operative program. As it regards the radon issue, several strategies have been implemented, including mapping, testing of homes, etc., with a large investment of efforts and human and financial resources, but the lack of a unique integrated methodology for management risk has so far led to the waste of resources and not yet to an achieved awareness among authorities and the general public about the related health risks.

In this paper, a comprehensive strategy and the specific activities for managing the radon issue in a practical and effective way have been proposed. The strengths of the proposed methodology are the practical tools proposed for the management of the radon potential from soil (through maps) and from building materials (through voluntary labels), as well as the remote control of indoor radon levels (through real-time sensor systems) and the integration of all these data in a radon certification for buildings.

The development of strategies and solutions is not regulated, but the proposed solution could be a starting point for a general harmonized methodology for long-term management plans. Providing public information and education on radon gas and potential remediation options is also an important first step to manage the social and behavioral factors, similar to human biomonitoring for the genetic and biological factors [34]. Further studies will focus on completing the methodology by modeling an integrated approach to manage the other determinants, by revising and optimizing public information and education programs to manage the social and behavioral factors, and by proposing new research on human biomonitoring for the control and monitoring of the genetic and biological factors.

## Figures and Tables

**Figure 1 ijerph-18-08089-f001:**
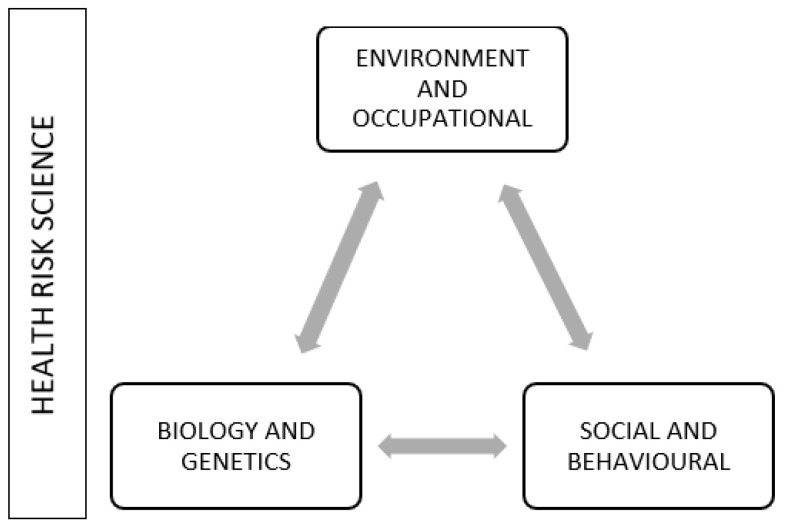
Three basic sets of health risk science. The arrows, in gray, indicate the mutual interaction among determinants.

**Figure 2 ijerph-18-08089-f002:**
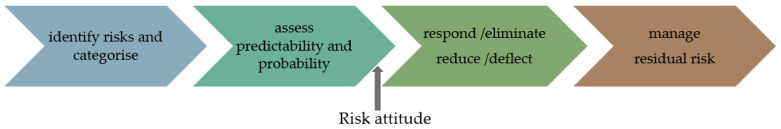
Logic workflow of the proposed strategy [20]. (Reproduced with permission from Mancini S., PhD thesis, 2018).

**Figure 3 ijerph-18-08089-f003:**
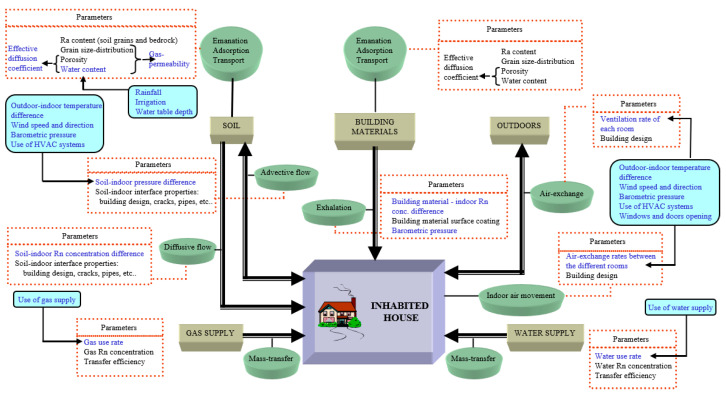
Diagram of the sources (brown square boxes), processes (green round boxes), and parameters that a global dynamic radon model must consider. The time-dependent parameters are in blue. Figure by Font L, available at https://icnts2008.bo.infn.it (accessed on 15 December 2021). Reprinted with permission from Lluis Font (2008).

**Figure 4 ijerph-18-08089-f004:**
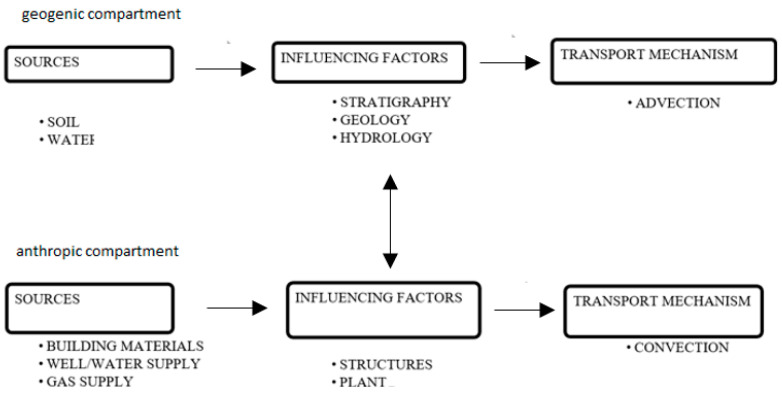
Environmental-occupational determinant: identifying risk and categorizing sources.

**Figure 5 ijerph-18-08089-f005:**
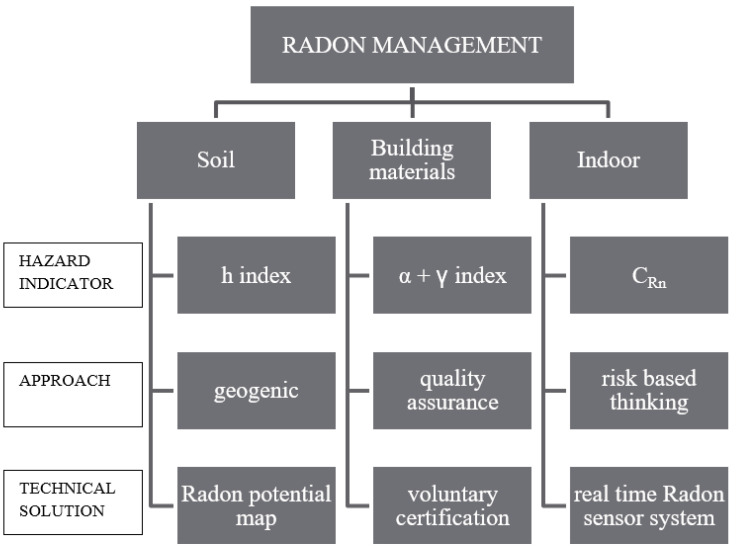
Logic scheme of the proposed strategy.

**Table 1 ijerph-18-08089-t001:** Assessment of predictability and probability. Selection of the main indexes, from the literature, for each source (water, building materials (BM), soil).

	Soil	BM	Water
Selected index	GRI ^1^	I_RP112Rn_ ^2^	C_w_ ^3^
Authorship	Cinelli G. et al. 2020 [24]	Trevisi et al. 2013 [23]	Nazaroff W. et al. 1987 [25]

^1^ Geogenic radon risk index; ^2^ Activity index for construction materials expressed in Bq/kg; ^3^ Radon activity concentration in water (C_w_) in Bq/L.

**Table 2 ijerph-18-08089-t002:** Assessment of predictability and probability. Analysis of pros and cons of the selected indexes.

	Soil GRI	BM I_RP112Rn_	Water C_w_
PROS	Geogenic approach. Technically simple. Accurate.	γ + α-exposure controlled.	Simple. Quite accurate.
CONS	Many input quantities. Predictive capacity not as expected.	Not internationally harmonized. E ^4^ measurement technique based on radium activity concentration measurements.	--

^4^ E = exhalation rate.

**Table 3 ijerph-18-08089-t003:** Proposed indexes.

	Soil	BM
index	GRC ^1^ [26]	γ + α
PROS	Geogenic approach. Technically simple. Few input parameters. Good predictive capacity.	Technically simple. E measurements technique based on radon activity concentrations.
CONS	Neglected γ radiation background.	Necessity to standardize E technique.

^1^ Geogenic RadCampania.

**Table 4 ijerph-18-08089-t004:** Soil index classes with indication of the acronym, level, descriptor by color, index value and required actions.

Class n.	Acronym	Level	Descriptor	Index Value	Action
0	S0	null		0	none
1	S1	very low		I_S_ ≤ 0.4	none
2	S2	low		0.4 < I_S_ ≤ 0.6	none or some
3	S3	medium		0.6 < I_S_ ≤ 0.8	some
4	S4	high		0.8 < I_S_ ≤ 1	almost 2
5	S5	very high		I_S_ > 1	more than 2

**Table 5 ijerph-18-08089-t005:** BM index classes with indication of the acronym, level, descriptor by color, index value and required restrictions for its use.

Class n.	Acronym	Level	Descriptor	Index value	Restrictions
0	BM0	null		0	None
1	BM1	very low		I_BM_ ≤ 0.4	None
2	BM2	low		0.4 < I_BM_ ≤ 0.6	None or some
3	BM3	medium		0.6 < I_BM_ ≤ 0.8	Some
4	BM4	high		0.8 < I_BM_ ≤ 1	Several
5	BM5	very high		I_BM_ > 1	Indoor use not recommended

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
