# Peer review of "A Novel Strategy for the Assessment of Radon Risk Based on Indicators"

_ijerph, 2021, doi:10.3390/ijerph18158089_

Round 1

Reviewer 1 Report

Dear Authors

I have accepted the answers

best regards

Reviewer 2 Report

The paper have been modified in detail and can be accepted.

Reviewer 3 Report

The authors have addressed my initial comments.

This manuscript is a resubmission of an earlier submission. The following is a list of the peer review reports and author responses from that submission.

Round 1

Reviewer 1 Report

The submitted manuscript entitled “Development of a methodological approach for the management of the health risk related to the exposure to Environmental Radon” describes and tried to review and explain a strategy and the specific activities to manage the radon issue in a practical and effective way from the dose assessment and its health risk point of view. Indeed, the author just reviewed and reported the results and finding of scientific communities (UNSCEAR, ICRP and WHO) and other surveys’ measurement, dosimetry and resulting doses exposure to radon, as well as highlighting a need to the methodology and guideline in order to support scientifically the public authorities in the pursuance of the protection of the citizens’ health at the high level of natural radioactivity.

I feel there is insufficient technical content and novelty to recommend publication for this journal. As a result, although the article does well written, the focus of the research is very narrow and obvious and whilst generally understandable. Besides that, the introduction does not set out the context, past research and novel contribution properly. The paper is also poor in terms of scientific discussion and state of the art of the work, relying on inappropriate references and inappropriate presentation of results.

 Below I have provided few remarks on the text:

  1. In the introduction part, the facts are pasted one after another without any obvious link and it is too long!! Also, it is recommended to rewrite it for better representation of information and cite the newest articles or documents.
  2. In the Tables, please define the abbreviations and acronyms. they should be defined the first time they are used in a table or figure.

Reviewer 2 Report

This paper proposed a methodological 19 approach for assessing the sustainability of environments concerning radon exposure. All claims in the submitted study are well described and technically sound. Therefore, the reviewer recommended the paper to be published after minor revision.

  1. For keywords, you could use different terms from the ones appearing in title and abstract in order to increase visibility at the moment of (future) search by readers.
  2. There are many conclusions in Section 4 (Conclusion), and you should add the discussion.
  3. Reference format should be uniform.
  4. In the research methods and data, the following references (with certain topics about health) should be added to the content and discussed in where applicable.

[1] Early warning and control of food safety risk using an improved AHC-RBF neural network integrating AHP-EW

[2] Static and dynamic energy structure analysis in the world for resource optimization using total factor productivity method based on slacks-based measure integrating data envelopment analysis.

Reviewer 3 Report

The manuscript illustrates a method for managing the risk of radon exposure in indoor environments., based on a tested conceptual framework. I recommend publishing this paper after some minor correction/revision.

  1. The primary concern is the transport mechanism of radon. The authors have not considered/discuss/highlight the importance of surface chemistry in this study. For instance, reactive surfaces may have the potential of prolonging the half-life of radon, and as well increase the health risk. In the Page 6, authors should explore further the transport mechanism, as the current mechanism does not assess the impact of particulates in transporting radon.
  2. Lines 50 - 65: Please include the drivers for the occurrence of radon in an indoor atmosphere. The source is well explained - but how does it get into (and trapped) in indoors; ventilation? Physisorption (and chemisorption) on active surfaces?
  3. The authors describe the mechanism of exposure in Lines 163-171. But it is necessary to include the transport mechanism.
  4. Authors should consider changing the title os Section 2 from "Material and Methods" to "Methods and Framework"
  5. Figure 1; If possible authors should be consistent in the use of "and" or "&". Also in Figure 2, please check why the "attitude" is marked red
  6. Authors should re-organise the Conclusion, and moved the excess information into the discussion aspect of the paper.
